# Intestinal Microbiota Differences in *Litopenaeus vannamei* Shrimp between Greenhouse and Aquaponic Rearing

**DOI:** 10.3390/life13020525

**Published:** 2023-02-14

**Authors:** Yabin Dou, Mengying Wen, Hui Shen, Sheng Zhang, Ge Jiang, Yi Qiao, Jie Cheng, Xiaohui Cao, Xihe Wan, Xiaoman Sun

**Affiliations:** 1College of Fisheries and Life Science, Shanghai Ocean University, Shanghai 201306, China; 2Marine Biology Lab, Jiangsu Marine Fisheries Research Institute, Nantong 226007, China; 3College of Food Science and Pharmaceutical Engineering, Nanjing Normal University, Nanjing 210023, China; 4School of Marine Science and Fisheries, Jiangsu Ocean University, Lianyungang 222005, China

**Keywords:** greenhouse rearing mode, aquaponic rearing mode, microbiota

## Abstract

The sustainability of shrimp aquaculture can be achieved through the development of greenhouse and aquaponic rearing modes, which are classified as heterotrophic and autotrophic bacterial aquaculture systems. However, there have been few investigations into the discrepancies between the intestinal and water microbiota of these two rearing methods. In this study, we collected shrimp samples from greenhouse-rearing (WG) and aquaponic-rearing (YG) ponds, and water samples (WE, YE), and investigated the intestinal and water microbiota between the two rearing modes. The results, through alpha and beta diversity analyses, reveal that there was basically no significant difference between shrimp intestine WG and YG (*p* > 0.05) or between rearing water WE and YE (*p* > 0.05). At the phylum and genus levels, the common bacteria between WE and WG differed significantly from those of YE and YG. The analysis of the top six phyla shows that Proteobacteria and Patescibacteria were significantly more abundant in the WG group than those in the YG group (*p* < 0.05). Conversely, Actinobacteriota, Firmicutes, and Verrucomicrobiota were significantly more abundant in the YG group than those in the WG group (*p* < 0.05). Venn analysis between WE and WG shows that *Amaricoccus*, *Micrococcales*, *Flavobacteriaceae*, and *Paracoccus* were the dominant bacteria genera, while *Acinetobacter*, *Demequina*, and *Rheinheimera* were the dominant bacteria genera between YE and YG. Pathways such as the biosynthesis of secondary metabolites, microbial metabolism in different environments, and carbon metabolism were significantly more upregulated in WG than those in YG (*p* < 0.05). In addition, pathways such as sulfate, chloroplast, phototrophy, and the nitrogen metabolism were significantly different between the WE and YE samples. These findings suggest that the greenhouse mode, a typical heterotrophic bacterial model, contains bacterial flora consisting of *Amaricoccus*, *Micrococcales*, *Flavobacteriaceae,* and other bacteria, which is indicative of the biological sludge process. Conversely, the aquaponic mode, an autotrophic bacterial model, was characterized by *Acinetobacter*, *Demequina*, *Rheinheimera,* and other bacteria, signifying the autotrophic biological process. This research provides an extensive understanding of heterotrophic and autotrophic bacterial aquaculture systems.

## 1. Introduction

The *Litopenaeus vannamei* species was first introduced to China in 1988 and has since experienced a surge in popularity. As the need for nutritious food rises, farmed shrimp has globally become the fastest-growing food product [1]. In 2016, China produced an estimated 1,672,246 tons of shrimp, as reported by the FAO. The expansion of shrimp aquaculture has had a detrimental effect on the environment due to wastewater. To mitigate this, circular farming models should be implemented to reduce nutrient content in aquatic environments and ensure sustainable aquacultural development. In Jiangsu, the current shrimp breeding mode is mainly greenhouse ponds that involve the periodic addition of probiotics and carbon into the ponds [2]. The greenhouse mode is characterized by lower water consumption and greater yield as compared to outdoor earth-pond mode. Unlike the greenhouse-enclosed raceway mode [3], Jiangsu’s greenhouse mode is combined with the earth pond and plastic warming shed, thus enabling the pond sediment to interact with the aquacultural system.

An aquaponic system is characterized by a symbiotic relationship among aquatic livestock, plants, and biofloc. Bacteria are responsible for transforming nitrogen-rich effluent into forms that can be absorbed by plants, thereby improving nutrient removal efficiency and producing vegetables [4]. In recent years, aquaponic systems between fish and vegetables have been a focus of research, particularly in terms of freshwater fish and vegetables [5]. However, the aquaponic system relationship between maricultural animals and plants has rarely been reported. Despite a few reports of the symbiosis between *Litopenaeus vannamei,* and gracilaria and sea anemone having been published in relation to seawater conditions [6,7], there is still no report regarding an aquaponic system between *Litopenaeus vannamei* and vegetables under brackish conditions (salinity 5–15). In this study, an aquaponic rearing mode between *Litopenaeus vannamei* and lycium barbarum is explored under the condition of salinity = 10, which has yielded successful breeding results.

To sum up, greenhouse and aquaponic rearing modes are mainly based on heterotrophic and autotrophic bacterial systems, respectively. Previous studies examined the dynamic characteristics of biofloc [8,9,10] and circulation modes [11,12] in shrimp aquaculture. It is commonly acknowledged that the microbiome of the host and its environment are pivotal to the host’s health and the stability of the environment [13]. However, there is a lack of understanding regarding the microbiota and the dissimilarity in microbiota between the two modes throughout the rearing period. Consequently, this study explores the differences in intestinal and water microbiota between the greenhouse and aquaponic rearing modes of the same batch of larvae. The results of this study could provide a better understanding of heterotrophic and autotrophic bacterial aquacultural systems.

## 2. Materials and Methods

### 2.1. Sample Collection

The samples of shrimp and water were collected from a commercial marine shrimp hatchery on 25 September 2021, with the shrimp having been bred together from larvae into juveniles (Figure 1). In greenhouse mode, density is fixed at 400 tails per square meter, whereas it is 1000 tails per square meter in aquaponic mode. After 30 days of breeding, the juveniles were transferred into greenhouse and aquaponic ponds. Shrimp individuals and six water samples were collected from the greenhouse (body length 12.7 ± 0.4 cm, *n* = 12) and aquaponic ponds (body length 11.9 ± 0.4 cm, *n* = 12) when juvenile shrimp had been bred for 60 days. The shrimp intestines were divided into two groups, one group collected from greenhouse ponds (WG) and one from aquaponic ponds (YG). Considering interindividual variations, three intestines from each group were combined to form a single microbial sample, while three hepatopancreases from the sample group were pooled for pathogen testing. The shrimp’s surface was sterilized with 70% alcohol, and the intestines were carefully dissected in the field and kept at −80 °C until further use. A total of 12 samples were collected for 454 instances of pyrosequencing and data processing, which were shrimp-intestine (WG, YG) and water (WE, YE) samples.

### 2.2. DNA Extraction

The shrimp-intestine, water-filter, and bacterial-colony samples were homogenized and blended with 1 mL of TE buffer (1 M Tris-HCl, 0.5 M EDTA, pH 8.0) and 0.3 g of sterilized quartz sand. This mixture was then processed on a FastPrep-24 Homogenization System (MP Biomedicals, Irvine, CA, USA) for three cycles of 45 s each at a speed of 6.0 m/s. Afterwards, the total DNA of the intestines, water filters, and bacterial colonies was extracted using PowerFecal DNA Isolation Kit and PowerWater DNA Isolation Kit (Mobio) in accordance with the manufacturer’s instructions.

### 2.3. Various Pathogens Test

To determine the presence or absence of different shrimp pathogens, including white-spot syndrome virus (WSSV), *Enterocytozoon hepatopenaei* (EHP), acute hepatopancreatic necrosis disease *Vibrios* (VP_AHPND_), and infectious hypodermal and hematopoietic necrosis virus (IHHNV), the purified DNA templates of all samples were tested using the primers as described [14]. Subsequently, the PCR products were subjected to electrophoresis on 1.5% agarose gels.

### 2.4. PCR Amplification

The V3–V4 regions of the bacterial 16S rRNA gene were amplified from 50 ng of DNA extracts for each sample using primer sets 341F and 806R. Three replicates of 16S amplification were produced for the DNA sample. The triplicate PCR products were combined using an AxyPrep DNA Gel Extraction Kit (Axygen, Hangzhou, China) and a Quant-iT PicoGreen double-stranded DNA assay (Invitrogen, Carlsbad, CA, USA). The amplicons from reaction mixture were purified, pooled at an equimolar ratio, and subjected to emulsion PCR to set up the amplicon libraries. Subsequently, sequencing was performed using a Roche Genome Sequencer FLX Titanium platform. The generated DNA sequences were analyzed with the QIIME toolkit, version 1.9.1 [15].

### 2.5. 454. Pyrosequencing and Data Processing

The Illumina sequencing data were analyzed using the Fast Length Adjustment of Short Reads (FLASH, vl.2.11) program to obtain the tags for merging pairs of reads [16]. These were clustered into operational taxonomic units (OTUs) with a cut-off value of 97% using UPARSE software v7.0.1090 [17]. The OTUs were compared with the Gold database, and any putative chimeras were removed using UCHIME v4.2.40 [18]. Alpha diversity indices such as Chao1 and Simpson were calculated to evaluate the community richness and diversity. Venn diagrams were generated using the R package to show unique and shared OTUs. Beta diversity analysis was conducted using the unweighted UniFrac distance metric to identify the relationships between different samples on the basis of the bacterial community [19], which was visualized via principal coordinate analysis (PCoA) and unweighted pair group method with arithmetic mean (UPGMA) clustering [15]. To predict the functions of the intestinal microbiota, the Kyoto Encyclopedia of Genes and Genomes (KEGG) database and PICRUSt software were used [20], and FaPROTAX was employed to predict the functions of the water microbiota (YE and WE). PCoA was performed with the use of the WGCNA, stat, and ggplot2 packages in R software (Version 2.15.3). To interpret the distance matrix, unweighted pair-group method with arithmetic means (UPGMA) clustering, a form of hierarchical clustering, was applied with average linkage using QIIME software (Version 1.7.0). The Wilcoxon rank-sum test was conducted to assess the differences between microbiome data, and those with a *p*-value of less than 0.05 were deemed statistically significant.

### 2.6. Statistical Analysis

All statistical analyses in this study were conducted using R package software. One-way analysis of variance (ANOVA) was applied to analyze Chao, ACE, Shannon, the relative abundance of phyla, and the relative abundance of genus data was established using a confidence level of 95%.

## 3. Results

### 3.1. Shrimp Physiological State and Pathogen Analysis

After 30 days, the same batches of larvae were divided into two groups. After a 60-day culture period, both groups of shrimp were in a good state of health. However, the aquaponic-reared group was evaluated as “good”, with approximately one kilogram of dead shrimp in the aquaponic pond every day. In contrast, the greenhouse-reared shrimp had no deaths and was evaluated as “very good”. The production of the greenhouse-reared shrimp was 2.5 kg/m^2^, while the production of the aquaponic-reared shrimp was 4 kg/m^2^. Furthermore, the body size of the greenhouse-reared shrimp was significantly larger than that of the aquaponic-reared shrimp (*p* < 0.05). Additionally, no pathogens such as EHP, VP_AHPND_, IHHNV, and WSSV were detected.

### 3.2. OTUs and Diversity of the Intestinal Microbiota

The average number of clean reads obtained from the shrimp intestines per sample was 32,156 ± 6591. The Chao1, ACE, and Shannon indices showed that the shrimp and water collected from the greenhouse pond were significantly higher than those collected from the aquaponic pond (*p* < 0.05) (Figure 2).

The Venn diagram reveals that Proteobacteria, Actinobacteriota, and Bacteroidota were among the 13 bacterial phyla shared by all four groups: WE, WG, YE, and YG (Appendix A). Moreover, 74 bacteria at the genus level were common among them, including *Amaricoccus*, *Micrococcales*, and *Paracoccus* (Appendix A). The common bacterial phyla between WE and WG were similar to those between YE and YG, with Proteobacteria and Actinobacteriota being the most prominent. However, the following bacterial phyla were markedly different (Figure 3A–D). The common phyla in WE and WG were Bacteriodota (11.32%), Patescibacteria (5.68%), Chloroflexi (1.96%), and others (2.03%). Meanwhile, the common phyla in YE and YG were Firmicutes (4.57%), Bacteriodota (3.47%), Verrucomicrobiota (1.38%), and others (1.62%). Additionally, the common genera of bacteria between WE and WG were distinct from those between YE and YG (Figure 3E–H). The most common genera in WE and WG were *Amaricoccus* (49.76%), *Micrococcales* (13.74%), *Flavobacteriaceae* (9.93%), and *Paracoccus* (6.72%); *Acinetobacter* (27.45%), *Demequina* (12.58%), *Rheinheimera* (10.92%), *Paracoccus* (6.87%), *Enterobacteriaceae* (6.17%), and *Rhodobacteraceae* (4.93%) were the most common genera in YE and YG.

PCoA shows that the shrimp intestines and water samples collected from the greenhouse ponds were similar but distinct from those of the aquaponic ponds. Additionally, the intestinal groups in the aquaponic ponds were dissimilar to those of aquaponic water (Figure 4). Nevertheless, there were no significant differences among WG, WE, YE, and YG (*p* > 0.05).

### 3.3. Community Composition of the Microbiota among Samples

The analysis of the dominant phyla and genera reveals that the relative abundances varied among the WG, YG, WE, and YE groups. The four shared phyla among these groups were Proteobacteria, Actinobacteria, Bacteroidetes, and Firmicutes. Proteobacteria and Patescibacteria were more abundant in the WG group compared to the YG group, while Actinobacteriota, Firmicutes, and Verrucomicrobiota were more abundant in the YG group than in the WG group (Figure 5a). Additionally, the relative abundances of *Amaricoccus*, *Micrococcales*, *Saccharinonadales*, *Rhodobacteraceae,* and *Demequina* at the genus level were significantly greater in the WG group than those in the YG group (Figure 5b). Differences were observed between the WE and YE groups, with the abundances of all top six phyla except Bacteroidota being higher in the WE group than those in the YE group (Figure 5c). The abundances of the top nine genera were also higher in the WE group than those in the YE group (Figure 5d). We found 15 significantly different genera among the 4 groups (Figure 6a). *Photobacterium* was mainly present in the YE and YG groups, and was highly abundant in the YG group (Figure 6g), although it did not show any statistically significant differences in abundance among the four groups.

### 3.4. Functional Prediction

The intestinal microbiota of WG and YG were annotated at KEGG Level 3 to identify their respective KEGG functions. Of the top 50 pathways, the majority were categorized as metabolic pathways, with 38 pathways exhibiting significant differences between WG and YG. Compared with the YG group, a number of pathways were significantly upregulated in the WG group, including the biosynthesis of secondary metabolites, microbial metabolism in diverse environments, and carbon metabolism. Conversely, several pathways, such as amino sugar and nucleotide sugar metabolism, alanine, aspartate, and glutamate metabolism, and sulfur metabolism, were upregulated in the YG group (Figure 7).

FaPROTAX annotation revealed that the water microbiota (WE vs. YE) had distinct functions. Pathways such as sulfate, chloroplasts, phototrophy, and nitrogen metabolism were significantly different between the WE and YE groups (Figure 8). Of the top 32 pathways, 19 were significantly different between WE and YE. The WE group had a higher abundance of pathways related to sulfur oxidation, methylotrophy, and phototrophic metabolism, whereas the YE group had a higher abundance of pathways related to phototrophy, fermentation, and chemoheterotrophy.

## 4. Discussion

There is a lack of understanding of the difference between greenhouse earth ponds and aquaponic ponds rearing *L. vannamei*, and we hypothesized that there were large differences between them. The analysis of alpha diversity showed that the microbial diversity of the water samples (WE, YE) was significantly higher than that of the intestinal samples (WG, YG). This suggests that the microflora in shrimp intestines was largely derived from the nearby aquatic environment or that the two contributed to the formation of microflora [14], and shrimp intestinal microbiota can act as a barrier against the surrounding environmental microbiota [21]. Despite the different breeding modes of the two systems, the Chao, Shannon, and ACE indices between the water and intestinal samples did not differ significantly, indicating that there was no considerable contrast in alpha diversity between the greenhouse, and aquaponic shrimp and water (Figure 2). Furthermore, the growth process of prawn in a fixed water body lacks essential minerals and trace elements, whereas the greenhouse model has reduced deficiency in this regard due to its contact with the soil.

At the phylum level, Venn analysis between WE and WG revealed that the common bacteria were, in descending order, Proteobacteria, Actinobacteriota, Bacteroidota, Patescibacteria, and Chloroflexi. These are different from the common bacteria between YE and YG: Proteobacteria, Actinobacteriota, Firmicutes, Bacteroidota, and Verrucomicrobiota. Previous studies demonstrated that Proteobacteria, Firmicutes, Actinobacteria, Cyanobacteria, Tenericutes, Planctomycetes, Bacteroidetes, Chloroflexi, and Verrucomicrobia are the dominant phyla in *L. vannamei* and the surrounding environment [22,23]. In this study, Proteobacteria and Actinobacteriota are the most prevalent bacteria in both culture modes. The abundance of Actinobacteriota is particularly remarkable.

Investigations revealed the presence of Patescibacteria in various environments, such as ground water sediments, activated sludge, and lakes [24], and demonstrated their involvement in the denitrification of nitrite [25]. In this study, Patescibacteria were common between WE and WG, with high abundance. This suggests that the water in the greenhouse pond had transformed into biofloc such as sludge with abundant bacterial diversity, since a certain amount of glucose was added to the greenhouse pond every day. Furthermore, Patescibacteria could have a parasitic relationship with Bacteroidota in partial-nitration/anammox reactors [26]. This study’s findings are in agreement with previous research suggesting that Patescibacteria may have an ecological function in furnishing lactate to other bacteria that inhabit the same environment [27].

The analysis of the Verrucomicrobiota phylum revealed that the *Luteolibacter* genus was the dominant species in the YE and YG groups, which is often found in recirculating aquacultural systems (RASs), wetlands, and activated sludge [28,29]. Additionally, the abundance of the Firmicutes phylum was significantly higher in the YE and YG groups than in the WE and WG groups, which was dominant in an RAS connected with a solid-phase denitrification reactor for nitrate and nitrite removal [30]. This suggests that Verrucomicrobiota and Firmicutes were the representative phyla of the YG and YE groups, indicating that they were the specific phyla of RAS.

At the genus level, Venn analysis between WE and WG revealed that *Amaricoccus*, *Micrococcales*, *Flavobacteriaceae*, and *Paracoccus* were the dominant bacterial genera, while *Acinetobacter*, *Demequina*, and *Rheinheimera* were the dominant bacterial genera between YE and YG. This indicates that the core flora between the greenhouse and aquaponic modes was significantly different. The *Amaricoccus* (Figure 5b), *Paracoccus* (Figure 5d), and *Tessaracoccus* bacterial genera were common between WE and WG, significantly higher than those between YE and YG. The *Amaricoccus* and *Paracoccus* genera are typical aerobic denitrifying bacteria that have the ability to decompose a broad range of organic compounds and transform nitrate into N_2_ [31]. This suggests that water containing sediment in the greenhouse pond had been transformed into activated sludge with the use of glucose and adequate aeration. In addition, OTU361 and the *Demequina* genus were both classified as Actinobacteria, with OTU361 being significantly more abundant in the WG group, and *Demequina* being significantly more abundant in the YG group. This class of bacteria was associated with antagonistic activity against the pathogenic *Vibrio* [32], suggesting that the two organisms had distinct characteristics between the two aquacultural modes. *Acinetobacter*, a significant contributor to hydroponics, increases plant biomass and eliminates nutrients [33]. It is commonly found in the rhizosphere of plants and has higher abundance when the levels of dissolved nutrients in the water are high [34]. In this study, *Acinetobacter* was the most abundant genus among the genera shared between YE and YG, accounting for 27.45%, while *Rhizobia* was the second most abundant, comprising 3.81% (Figure 3H). This result is in agreement with previous studies on the microbiota of hydroponic systems. However, the *Acinetobacter* genus may be responsible for red leg disease in *L. vannamei* in shrimp [35]. *Acinetobacter*, as a key bacterium in the aquaponic system, may have a twofold effect considering the fact that there was a kilogram of prawns per day in the aquaponic mode.

The YG group had higher abundances of *Acinetobacter* (Figure 5c) and *Photobacterium* (Figure 5g) than those of the WG group. These genera are opportunistic pathogens. The *Photobacterium* species are commonly found in marine environments and were linked to the pathogenesis of aquatic organisms, such as shrimp and fish [36,37]. Additionally, the presence of *Rhizobium*, a symbiotic bacterium typically found in plant roots [38] in the YG group suggests that the shrimp intestinal microbiota interacted with bacteria from the plant roots.

Previous studies indicated a strong relationship between the microbiota of shrimp intestines and their surrounding water, suggesting that microorganisms from ambient water can interact with the microbiota of aquatic animals [39,40]. In this study, 20 common phyla were identified between WE and WG, and there were no unique phyla in WG compared to WE (Figure 3A). In contrast, 8 unique phyla were observed in YG compared to YE (Figure 3B), suggesting that the interaction between the shrimp intestines and water in greenhouse mode was stronger than that in aquaponic mode. It is likely that the presence of microflocs in the water body of the greenhouse mode is what caused the strong shaping effect on the shrimp intestines as they fed on these microflocs. The intestinal microbiota of healthy shrimp may be more dependable than that of the surrounding water, as it creates a microenvironment that encourages the growth of certain microorganisms [21]. In the aquaponic mode, water is filtered through a microfilter, resulting in a lack of microflocs, and thus minimal interaction between shrimp intestines and the microbial flora of the water body. This suggests that, to mediate the interaction between shrimp intestines and water microbiota, microflocs can be used.

PCoA results revealed that the correlation between the WE and WG groups was greater than that between the YE and YG groups (Figure 4), suggesting that the shrimp grown in the greenhouse pond were in better health than those grown in the aquaponic pond. Notably, the YG group was distinct from the WG and WE groups, even though they were together for the initial 30 days, indicating that there were considerable differences between the greenhouse and aquaponic modes. It is evident that the intestinal flora of shrimp is largely influenced by the aquatic environment flora. Additionally, the Venn diagram indicates that the number of bacterial varieties in shrimp intestines and water bodies in the greenhouse mode was lower than that in the aquaponic mode at the phylum and genus levels (Figure 3). We hypothesize that the addition of carbon sources in the greenhouse mode could lead to an increase in the abundance of bacteria with high C/N, thus reducing the species of bacteria with low C/N.

The microbiome of greenhouse rearing and aquaponic rearing modes showed close correlation with the taxa. Among the top 50 abundant pathways, metabolic pathways such as carbon metabolism, the biosynthesis of secondary metabolites, purine metabolism, glyoxylate and discarboxylate metabolism, pyruvate metabolism, pyrimidine metabolism, tryptophan metabolism, glutathione metabolism, and histidine metabolism were more abundant in the WG group than those in the YG group, indicating that a higher proportion of heterotrophic biomass, and better digestion and absorption of nutrients [25]. However, some pathways such as alanine, aspartate, and glutamate metabolism were significantly lower in the WG group than those in the YG group. These pathways were upregulated by environmental stress, such as exposure to microcystin-LR [41] and low long-term salinity [42]. Although the WG and YG groups had the same salinity of 10 and had not been exposed to microcystin-LR, the responses of intestinal function were very different between them. This suggests that the ecosystem of the greenhouse system was more favorable than that of the aquaponic system, and that the shrimp reared in aquaponic ponds were under some unknown stress. Interestingly, the YG group exhibited a significantly greater sulfur metabolism than that of the WG group, which was likely due to the addition of ammonium sulfate to the YG group to form biofilms of autotrophic denitrifying bacteria.

The differences between the predicted functionalities of the microbiome in the WG and YE groups were more significant than those between the WG and YG groups. According to the annotation of FaPROTAX, the YE group had significantly higher levels of photoheterotrophy, phototrophy, hydrocarbon degradation, thiosulfate respiration, and cellulolysis than those of the WE group, suggesting that these results are consistent with those of a aquaponic system that combined plants and biological floating biomass [43]. Previous studies found that methylotrophy is largely driven by the orders of *Rhodobacterales* and *Rhodospirillales* [44]. The WE group, which possessed higher light intensity than that of the YE group, displayed a significantly higher relative abundance of *Rhodobacterales* due to the higher levels of methylotrophy present. This process followed the same principles, as chloroplasts were more abundant than microalgae. Despite the considerable distinctions between the greenhouse and aquaponic systems, nitrogen cycles such as denitrification, nitrous oxide denitrification, nitrite denitrification, nitrate denitrification, nitrite respiration, and nitrate respiration did not show significant variations. This suggests that the autotrophic and heterotrophic denitrification systems in the YE and WE groups were capable of effectively removing water nitrogen.

## 5. Conclusions

To find the differences of microbiota rearing between greenhouse and aquaponic modes, we compared the microbiota of shrimp samples from greenhouse-rearing (WG) and aquaponic-rearing (YG) ponds, and water samples (WE, YE). Our results revealed that there were basically no significant differences in alpha and beta diversities between the two modes in terms of shrimp intestinal microbiota and water microbiota. However, the common bacteria between WE and WG differed significantly from those of YE and YG when analyzed at the phylum and genus levels. Pathways including the biosynthesis of secondary metabolites, microbial metabolism, and carbon metabolism were significantly more activated in WG than in YG (*p* < 0.05). Results suggest that the greenhouse model had bacterial flora composed of *Amaricoccus*, *Micrococcales*, *Flavobacteriaceae*, and other bacteria, indicating that it was a product of the biological sludge process. On the other hand, the aquaponic model, an autotrophic bacterial model, was characterized by *Acinetobacter*, *Demequina*, *Rheinheimera,* and other bacteria, signifying the autotrophic biological process.

## Figures and Tables

**Figure 1 life-13-00525-f001:**
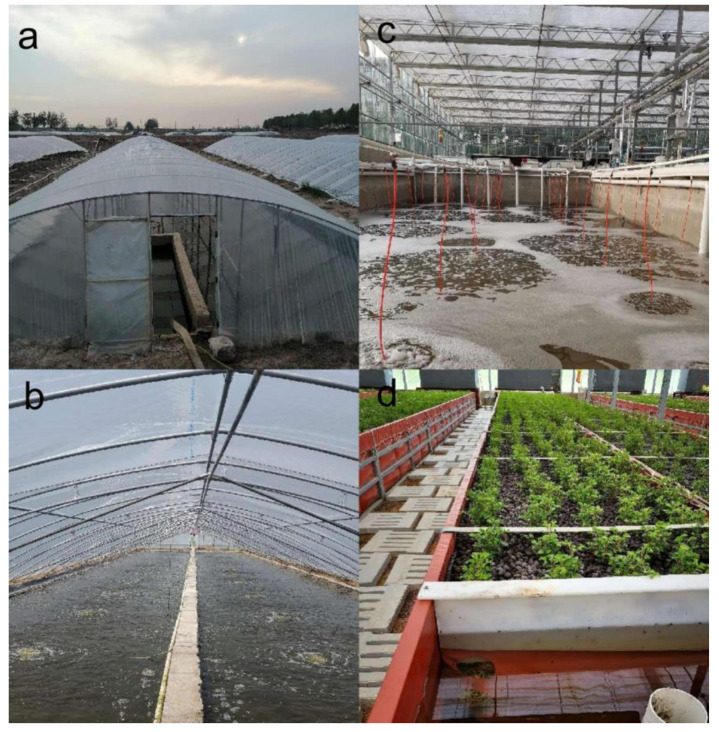
Greenhouse rearing mode and aquaponic rearing mode. (**a**) The external structure of the greenhouse ponds; (**b**) internal structure of greenhouse ponds with about 0.8 m depth water; (**c**) rearing pond of aquaponic rearing mode; (**d**) recyclable biobased packages combined with plants of aquaponic rearing mode.

**Figure 2 life-13-00525-f002:**
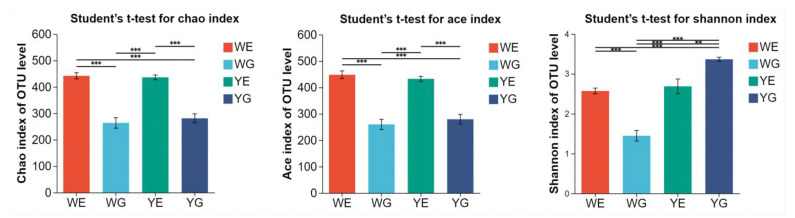
Richness and diversity of bacterial species in four groups. **, significant differences at 0.001 < *p* ≤ 0.01; ***, significant differences at *p* ≤ 0.001.

**Figure 3 life-13-00525-f003:**
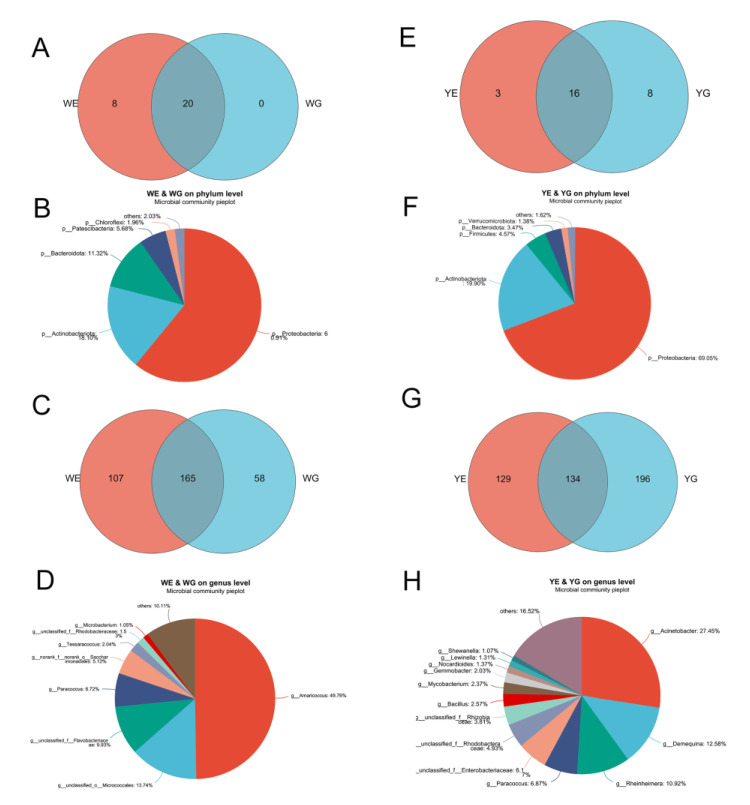
Venn diagrams among WE, WG, YE, YG, and the pie charts of the common bacteria. (**A**) Venn diagram between WE and WG at the phylum level; (**B**) pie chart of common phyla between WE and WG; (**C**) Venn diagram between WE and WG at the genus level; (**D**) pie chart of common genera between WE and WG; (**E**) Venn diagram between YE and YG at the phylum level; (**F**) pie chart of common phyla between YE and YG; (**G**) Venn diagram between YE and YG at the genus level; (**H**) pie chart of common genera between YE and YG.

**Figure 4 life-13-00525-f004:**
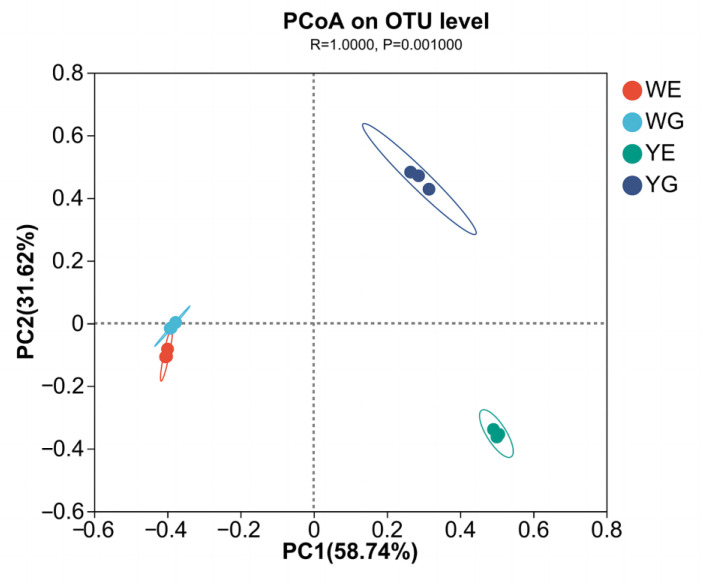
Principal coordinate analysis (PCoA) of bacterial communities.

**Figure 5 life-13-00525-f005:**
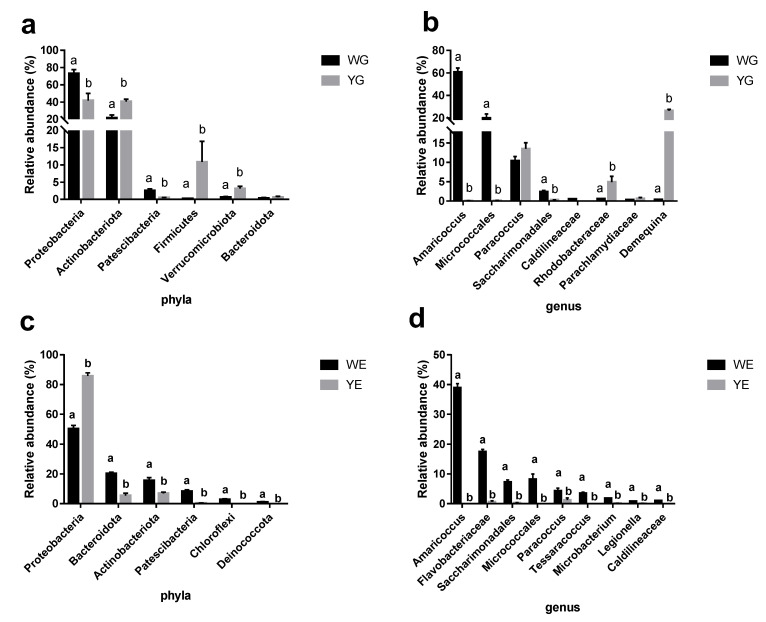
Significant difference analysis of the (**a**) top six most abundant phyla and (**b**) top eight most abundant genera between the WG and YG groups. Significant difference analysis of the (**c**) top six most abundant phyla and (**d**) top nine most abundant genera between the WE and YE groups. ^a, b^ The ranges indicated with another superscript are significantly different from each other (based on average ± standard deviation) according to a One-way ANOVA (*p* < 0.05).

**Figure 6 life-13-00525-f006:**
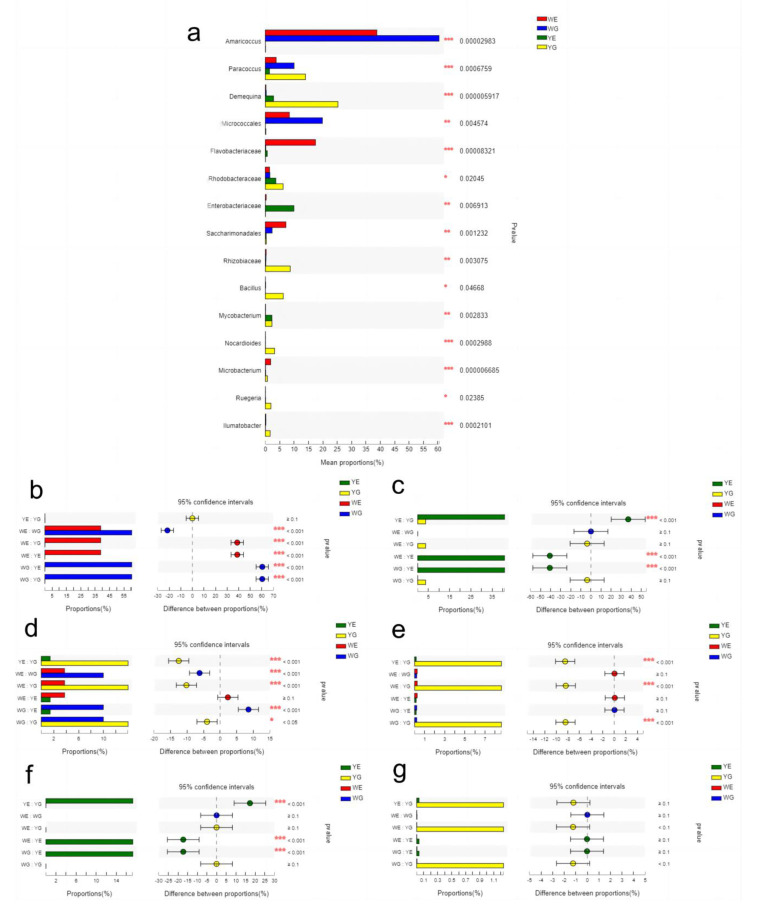
Comparison of bacterial abundances at the genus level among WG, WE, YG and YE. (**a**) One-way ANOVA bar plot at the genus level for (**b**) *Amaricoccus*, (**c**) *Acinetobacter*, (**d**) *Paracoccus*, (**e**) *Rhizobium*, (**f**) *Rheinheimera*, (**g**) *Photobacterium*. *, significant differences at 0.01 < *p* ≤ 0.05; **, significant differences at 0.001 < *p* ≤ 0.01; ***, significant differences at *p* ≤ 0.001.

**Figure 7 life-13-00525-f007:**
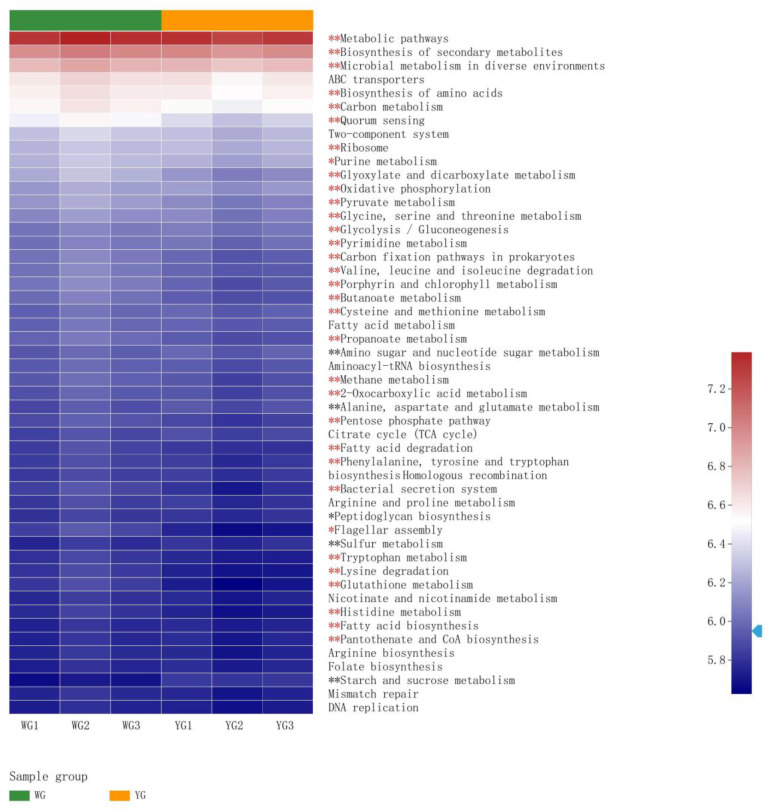
A Heat map and hierarchical clustering of significantly different KEGG functions (Level 3) between WG and YG. Red asterisks, significantly upregulated pathways (WG vs. YG); black asterisks, significantly downregulated pathways (WG vs. YG); * represents significant differences 0.01< *p* ≤0.05; **, significant differences at 0.001 < *p* ≤ 0.01; significant differences at *p* ≤ 0.001.

**Figure 8 life-13-00525-f008:**
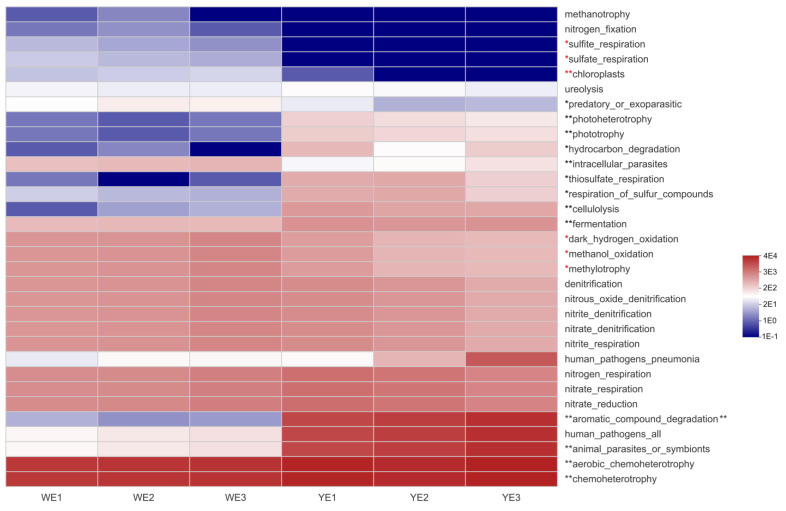
Visualization of significantly different FaPROTAX functions between WE and YE in the form of a heatmap and hierarchical clustering. Red asterisks, significantly upregulated pathways (WE vs. YE); black asterisks, significantly downregulated pathways (WE vs. YE); *, significant differences at 0.01 < *p* ≤ 0.05; **, significant differences at 0.001 < *p* ≤ 0.01.

## Data Availability

Data are provided in the article.

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
