# Peer review of "Intestinal Microbiota Differences in Litopenaeus vannamei Shrimp between Greenhouse and Aquaponic Rearing"

_life, 2023, doi:10.3390/life13020525_

Round 1

Reviewer 1 Report

Three rearing modes are open pond, greenhouse and intensive aquaculture (named 'factory rearing'). In my opinion the term 'factory rearing' could be replaced with either intensive aquaculture or aquaponic because its treatment is connected with plant units. However, intensive aquaculture is sufficient where the plant unit could be considered simply as water treatment unit.

Figure represented are too small, fonts are not readable even after 400% zoom. Please rearrange your figure and split according to scope for example, between WG & WE also YG & YE. Do axis frame format with 0 horizontal and vertical gaps. This includes bar chart, venn diagram and other charts.

Author Response

Dear Editor,

Thank you very much for giving us an opportunity to revise our manuscript. We read with great interest the comments from reviewers. We are grateful to their comments.

According to the reviewers’ detailed suggestions, we have made a careful revision on the original manuscript.

Reviewer 1

  1. Three rearing modes are open pond, greenhouse and intensive aquaculture (named 'factory rearing'). In my opinion the term 'factory rearing' could be replaced with either intensive aquaculture or aquaponic because its treatment is connected with plant units. However, intensive aquaculture is sufficient where the plant unit could be considered simply as water treatment unit.

We agree with the reviewer, and we adjust the “factory rearing” to “aquaponic rearing”. Moreover, we added the introduction about aquaponic progress in aquaculture.  

  1. Figure represented are too small, fonts are not readable even after 400% zoom. Please rearrange your figure and split according to scope for example, between WG & WE also YG & YE. Do axis frame format with 0 horizontal and vertical gaps. This includes bar chart, venn diagram and other charts.

We have optimized all the figures, including the image quality and plate layout.

Reviewer 2 Report

Dear Author,

The manuscript has a good subject and is presented more understandable. I would suggest some minor points, especially stressing out typological usage. I would also suggest excluding references from the supplementary table and transferring the references to the manuscript file. The points I remarked on could be found in the attached document. 

Author Response

Response to Reviewers

Dear Editor,

Thank you very much for giving us an opportunity to revise our manuscript. We read with great interest the comments from reviewers. We are grateful to their comments.

According to the reviewers’ detailed suggestions, we have made a careful revision on the original manuscript.

Reviewer 2

The manuscript has a good subject and is presented more understandable. I would suggest some minor points, especially stressing out typological usage. I would also suggest excluding references from the supplementary table and transferring the references to the manuscript file. The points I remarked on could be found in the attached document. 

We are deeply appreciative of the reviewer for positive feedback on this article. We have amended the minor errors indicated in the author’s article individually. Please refer to the revised manuscript.

Reviewer 3 Report

Aims to explore the differences in the intestinal and water microbiota between the greenhouse and factory rearing modes of the same batch of larvae. Authors collected shrimp samples from greenhouse-rearing and factory-rearing ponds, and water samples, and compared the intestinal and water microbiota between the two rearing modes. Results suggested that the greenhouse model, a typical heterotrophic bacteria model, contains a bacterial flora consisting of Amaricoccus, Micrococcales, Flavobacteriaceae and other bacteria. This research provides an extensive understanding of heterotrophic and autotrophic bacterial aquaculture systems. There are some commnets as below:

1. the introduction is too simple to understand the purpose of this study. Please detail describe the research background in this part.

2. L88, six water samples including greenhouse and factory?

3. L140-144, please provide the packages name

4. the results part is good, but too long to read.

5. In figure 4, there are only 12 samples, please introduce them in the method part.

6. the discussion is better.

Author Response

Response to Reviewers

Dear Editor,

Thank you very much for giving us an opportunity to revise our manuscript. We read with great interest the comments from reviewers. We are grateful to their comments.

According to the reviewers’ detailed suggestions, we have made a careful revision on the original manuscript.

Reviewer 3

Aims to explore the differences in the intestinal and water microbiota between the greenhouse and factory rearing modes of the same batch of larvae. Authors collected shrimp samples from greenhouse-rearing and factory-rearing ponds, and water samples, and compared the intestinal and water microbiota between the two rearing modes. Results suggested that the greenhouse model, a typical heterotrophic bacteria model, contains a bacterial flora consisting of Amaricoccus, Micrococcales, Flavobacteriaceae and other bacteria. This research provides an extensive understanding of heterotrophic and autotrophic bacterial aquaculture systems. There are some commnets as below:

In response to the reviewer’s feedback on the necessity to upgrade the English writing level of this article, we have made comprehensive modifications to the writing aspect of it. The edited section is highlighted in the revisions mode, please refer to the revised manuscript.

  1. the introduction is too simple to understand the purpose of this study. Please detail describe the research background in this part.

Based on reviewer’s opinion, the “factory rearing” is not suitable in this paper. We adjusted “factory rearing” to “aquaponic rearing”. Thus, we have augmented the research progress about the greenhouse rearing and aquaponic rearing modes. Please refer to the revised manuscript.

  1. L88, six water samples including greenhouse and factory?

Shrimp individuals and six water samples were collected from the greenhouse (body length 12.7 ± 0.4 cm, n=12) and aquaponic ponds (body length 11.9 ± 0.4 cm, n=12) when juvenile shrimp were bred for 60 days. The shrimp intestines were divided into two groups, the group collected from greenhouse ponds (WG) and aquaponic ponds (YG). Considering the inter-individual variations, three intestines from each group were combined to form a single microbial sample, while three hepatopancreases from the sample group were pooled for pathogen testing.

Yes, six water samples includes the greenhouse ponds and aquaponic ponds.

  1. L140-144, please provide the packages name

PCoA analysis was performed with the help of WGCNA, stat, and ggplot2 packages in R software (Version 2.15.3). To interpret the distance matrix, Unweighted Pair-group Method with Arithmetic (UPGMA) Means Clustering, a form of hierarchical clustering, was applied with average linkage, using QIIME software (Version 1.7.0).

  1. the results part is good, but too long to read.

In the results part, we displayed the Alphy, Beta diversity, the difference of phyla and genera between intestinal and water samples, the KEGG of intestinal microbiota and FaPROTAX annotation of water microbiota. The research results are certainly abundant, yet this is the first time to report the distinctions in the flora of the greenhouse and aquaponic, thus I would like to illustrate the results more, and I am sure you can comprehend. 

  1. In figure 4, there are only 12 samples, please introduce them in the method part.

A total of 12 samples were conducted for 454 Pyrosequencing and data processing, which were shrimp intestine (WG, YG) and water samples (WE, YE).

  1. the discussion is better.

We have modified some sentences in discussion section in order to render the discussion more logical and coherent. Please refer to revised manuscript.

Round 2

Reviewer 3 Report

this version is better. There is no comment.